

# Surviving without oxygen involves major tissue specific changes in the proteome of crucian carp (*Carassius carassius*)

Anette Johansen[1], Bernd Thiede[1], Jan Haug Anonsen[1,2] and Göran E. Nilsson[1]

[1] Department of Biosciences, University of Oslo, Oslo, Norway
[2] Climate & Environment Department, NORCE, Norwegian Research Centre AS, Stavanger, Norway

## ABSTRACT

The crucian carp (*Carassius carassius*) can survive complete oxygen depletion (anoxia) for several months at low temperatures, making it an excellent model for studying molecular adaptations to anoxia. Still, little is known about how its global proteome responds to anoxia and reoxygenation. By applying mass spectrometry-based proteome analyses on brain, heart and liver tissue from crucian carp exposed to normoxia, five days anoxia, and reoxygenation, we found major changes in particularly cardiac and hepatic protein levels in response to anoxia and reoxygenation. These included tissue-specific differences in mitochondrial proteins involved in aerobic respiration and mitochondrial membrane integrity. Enzymes in the electron transport system (ETS) decreased in heart and increased massively in liver during anoxia and reoxygenation but did not change in the brain. Importantly, the data support a special role for the liver in succinate handling upon reoxygenation, as suggested by a drastic increase of components of the ETS and uncoupling protein 2, which could allow for succinate metabolism without excessive formation of reactive oxygen species (ROS). Also during reoxygenation, the levels of proteins involved in the cristae junction organization of the mitochondria changed in the heart, possibly functioning to suppress ROS formation. Furthermore, proteins involved in immune (complement) system activation changed in the anoxic heart compared to normoxic controls. The results emphasize that responses to anoxia are highly tissue-specific and related to organ function.

# INTRODUCTION

The cyprinid freshwater fish crucian carp (*Carassius carassius)* and its close relative the goldfish (*Carassius auratus*) are among very few vertebrates that have evolved mechanisms allowing survival during complete oxygen depletion (*i.e.,* anoxia) for days, and in the case of crucian carp, for several months when temperatures are low (*Van den Thillart, Van Berge-Henegouwen & Kesbeke, 1983*; *Lutz, Nilsson & Peréz-Pinzón, 1996*). The small ponds they inhabit are covered by ice and snow during wintertime, blocking photosynthesis and oxygen diffusion from the air, eventually leaving the water anoxic (*Nilsson & Renshaw, 2004*). By a combination of reduced ATP demand (*Van Waversveld, Addink & Van den*

Corresponding author
Anette Johansen,
anette.johansen@medisin.uio.no

*Thillart, 1989*; *Johansson, Nilsson & Tornblom, 1995*; *Nilsson, 2001*) and increased glycolytic rate fueled by a large hepatic glycogen store (*Vornanen & Haverinen, 2016*), crucian carp can survive long-term anoxia and still avoid loss of energy charge. Cardiac output is maintained during anoxia (*Stecyk et al., 2004*) which ensures transport of glucose and anaerobic end products. *Carassius* also possess a unique mechanism where in red muscles, a modified anoxia-activated pyruvate decarboxylase and alcohol dehydrogenase convert lactate to acetaldehyde and further to ethanol, the latter excreted over the gills (*Nilsson, 1988*; *Fagernes et al., 2017*).

Protein synthesis is a highly energy-requiring process that can consume 60–90% of the ATP demand in some cells (*Pannevis & Houlihan, 1992*; *Smith & Houlihan, 1995*; *Wieser & Krumschnabel, 2001*). In crucian carp, protein synthesis in the heart and liver is reduced during anoxia, but the overall protein synthesis in the brain is unaffected or possibly only slightly suppressed (*Smith et al., 1996*). A study using 2-D gel electrophoresis to examine the crucian carp brain proteome revealed that the levels of a few selected proteins are indeed changed in anoxia (*Smith et al., 2009*). However, only a small selection of relatively abundant proteins was investigated, so there is a need for a more comprehensive approach to quantify the overall proteome changes that occur in the brain when oxygen levels are altered.

To date, there is only one report using a discovery-driven proteomic approach on the *Carassius* heart, showing that aldolases may contribute to a balance between glycolysis and gluconeogenesis during hypoxia (*Imbrogno et al., 2019*). Since this was a qualitative study performed on hypoxia-exposed goldfish, it is still not well understood how the cardiac proteome responds to anoxia and reoxygenation in the crucian carp. Most studies on anoxia-tolerant vertebrates have focused on the highly energy-demanding brain and heart (*Lutz & Nilsson, 2004*; *Farrell & Stecyk, 2007*), whereas the glucose-supplying liver is rarely in focus, despite its crucial importance for long-term anoxic survival.

The prominent role of succinate accumulation in mammalian ischemia-reperfusion (*Chouchani et al., 2016*) and anoxia-tolerant turtles (*Bundgaard et al., 2019*; *Fago, 2022*) has recently gained a lot of attention. Ischemic (and anoxic) succinate accumulation has been proposed to occur through reverse TCA activity, in which fumarate is reduced by a reverse-functioning succinate dehydrogenase (*Chouchani et al., 2014*), which would allow proton pumping by Complex I to maintain mitochondrial membrane potential and possibly ATP production. Others have proposed that ischemic succinate generation occurs primarily through succinate-CoA ligase activity (*Zhang et al., 2018*), which leads to ATP production *via* substrate level phosphorylation. High succinate levels have been ascribed a role in reperfusion damage due to insufficient capacity of the electron transport system (ETS) to channel the electrons produced leading to increased ROS generation at Complex I. This process, coined reverse electron transfer (RET), occurs once oxygen supply is resumed and succinate is oxidized to fumarate (*Murphy, 2009*; *Chouchani et al., 2014*). In a recent metabolomic study we found that succinate accumulated substantially in several tissues of the crucian carp during anoxia, but much more so in the liver (reaching 2,600 µM after 5 days) than in brain and heart (there reaching 500 and 650 µM, respectively) (*Dahl et al., 2021*). We also found that succinate was present at higher levels in anoxic blood

plasma (1,000 µM) than in brain and heart, which made us conclude that the liver is accumulating succinate transported in blood from other tissues (*Dahl et al., 2021*). This succinate transport to the liver tissue could potentially limit the production of ROS during reoxygenation in other tissues such as the heart and the brain, and we hypothesize that the succinate metabolism might be tissue-specifically regulated at the proteome level.

In the metabolomic study we also found several tissue-specific adaptations to anoxia and reoxygenation, among them a slower energetic recovery of the heart during reoxygenation compared to the brain and liver (*Dahl et al., 2021*). To elucidate the tissue-specific adaptations to anoxia and reoxygenation in the crucian carp, the aim of the present study was therefore to examine how the overall proteomes in the brain, heart and liver are affected by the removal and return of oxygen.

# MATERIALS & METHODS

## Materials
Acetonitrile (MS grade), acetone (HPLC grade), and water (HPLC grade) were purchased from VWR, Oslo, Norway. Ammonium bicarbonate, formic acid, iodoacetamide, urea and 1,4-dithiotreitol (DTT) were from Sigma-Aldrich, Oslo, Norway.

## Animal handling
Crucian carp were collected in the autumn from the Tjernsrud pond near Oslo, Norway, using nylon net cages. Fish were held at the aquarium facility at the Department of Biosciences, University of Oslo, in tanks with a semi-closed system supplied with dechlorinated, aerated Oslo tap water in a room with a 12:12 h light:dark cycle. The water temperature was kept at 10–12 °C. The fish were fed daily with commercial carp food (Tetrapond, Tetra, Melle, Germany), and acclimatized to indoor conditions for at least four weeks prior to exposure. All experimental procedures on animals in this project were approved by the Norwegian Animal Health Authority (approval no FOTS 16063), thereby following all relevant Norwegian and European Union guidelines and regulations.

## Anoxia exposure and tissue sampling
Prior to anoxia exposure, randomly selected crucian carp of both sex ($n = 18$, weight $= 27 \pm 8$ g) were transferred to separate, 25-liter lightproof containers, starved and acclimatized for 48 h in aerated water flowing through the tanks. In total 12 fish ($n = 6$ in the anoxic group and $n = 6$ in the reoxygenation group) were exposed to anoxia for five days. Sample size was decided based on previous studies (*Stensløkken et al., 2008*; *Dahl et al., 2021*). Anoxia was achieved by continuous low bubbling of nitrogen, and the oxygen level monitored using a Firesting fiber-optic oxygen meter with an oxygen probe (PyroScience GmbH, Aachen, Germany) to ensure that oxygen levels never exceeded 0.1% of air saturation ($<0.01$ mg/l $O_2$). Fish in the normoxic control group ($n = 6$) were treated identically but with aerated water only. Fish in the reoxygenation group ($n = 6$) were first kept anoxic for five days followed by one day of reoxygenation. The water temperature was maintained between 8.2–8.6 °C throughout the experiment. Tissue samples were collected after five days of anoxia; five days of normoxia and five days anoxia followed by one day

reoxygenation. Fish were euthanized by a quick blow to the head, the spinal cord was cut and brain, heart and liver tissues were dissected out in the mentioned order. The tissues were blotted dry, but no rinsed before they were snap-frozen in liquid nitrogen and stored at −80 °C until further analysis. All fish used in the present study were from the same experimental exposure. Crucian carp can withstand at least two weeks of anoxic exposure at 8 °C (*Nilsson, 1990*) and no mortality was observed during the experiment.

## Protein extraction and digestion

Frozen crucian carp brain, heart and liver tissues from six fish per experimental group were lysed in ice-cold SILAC$^{TM}$ Phosphoprotein Lysis buffer (Invitrogen, Carlsbad, CA, USA). Brain and liver tissues were homogenized with a pestle, whereas heart tissue was lysed with a Tissuelyser II instrument (Qiagen, Hilden, Germany) operating at 15 strokes/s for 2 min twice with a tungsten carbide bead. After 5–10 min incubation on ice and then at −80 °C, the buffer volume was adjusted to 60 mg tissue/mL (for brain and liver) or 25 mg tissue/mL (for heart) lysis buffer and the lysate cleared by centrifugation (18,000 g for 15 min). Total protein content was measured with a Detergent Compatible Bradford Assay Reagent (Pierce, Rockford, IL, USA) at 595 nm or the BCA assay (Pierce, Rockford, IL, USA) at 570 nm, both with BSA as a standard. From each biological replicate, 60 μg protein was precipitated with five volumes ice-cold acetone at −20 °C overnight. Precipitated protein was pelleted by centrifugation (13,000 g for 15 min), the acetone was aspirated and the pellet let to air dry before resuspension in 6 M urea in 100 mM ammonium bicarbonate. Cystines present in the sample were reduced with 10 mM DTT at 30 °C for 60 min and alkylated with 30 mM iodoacetamide at 22 °C for 1 h in the dark. The reaction was quenched with 30 mM DTT at 30 °C for 30 min and the sample diluted with 50 mM ammonium bicarbonate before digestion with 1 μg trypsin Mass Spec Grade (Promega, Madison, WI, USA) at 37 °C overnight. Finally, the digest was quenched with 1% formic acid and the peptides cleaned by solid-phase extraction (SPE) using a Ziptip-C18 (Millipore, Billerica, MA, USA).

## Nanoflow LC-MS/MS analysis

The peptides were analyzed using an Ultimate 3000 nano-UHPLC system (Dionex, Sunnyvale, CA, USA) connected to a Q Exactive mass spectrometer (ThermoElectron, Bremen, Germany) equipped with a nano electrospray ion source. Peptides were separated by liquid chromatography using an Acclaim PepMap 100 column (C18, 3 μm beads, 100 Å, 75 μm inner diameter, 50 cm; Dionex, Sunnyvale CA, USA). A flow rate of 300 nL/min was employed using 0.1% formic acid (solvent A) and 0.1% formic acid/90% acetonitrile (solvent B). A solvent gradient of 4–35% B in 160 min, to 50% B in 20 min and then to 80% B in 2 min was used. The mass spectrometer was operated in data-dependent mode to automatically isolate and fragment multiple charged precursors (top 10), and target ions already selected for MS/MS were dynamically excluded for 60 s. Survey full-scan MS spectra (ranging from m/z 300 to 2,000) were acquired with the resolution $R = 70,000$ at m/z 200, after accumulation to a target of 1E6. The method applied a maximum allowed ion accumulation time of 100 ms. Higher-energy collision induced dissociation (HCD) fragmentation was applied with a target value of 10,000 charges and a resolution $R = 17,500$

with NCE 28. The isolation window was $m/z = 2$ without offset and the maximum allowed ion accumulation for the MS2 was set to 60 ms. The lock mass option was enabled in MS mode for internal recalibration during the analysis and accurate mass measurements.

### Database search and label-free quantitation

Data were acquired using Xcalibur v2.5.5. Database searches were performed against the *Carassius auratus* database (NCBI; all taxons; 96703 entries), with PEAKS X+ software version 10.5 (Bioinformatics Solutions, Waterloo, ON, Canada). Digestion enzyme trypsin with one maximum missed cleavage, parent ion error tolerance of 10.0 ppm and fragment ion mass error tolerance of 0.1 Da were used in the database search. Furthermore, carbamidomethylation as fixed modification and oxidation of methionine and acetylation of the protein N-terminus were specified as variable modifications. The search was performed with a maximum number of two post-translational modifications (PTMs). The four (liver) or five (brain, heart) biological replicates that contained the highest and most comparable total intensities were selected to ensure robust label-free quantitation (LFQ). The following filtration parameters were applied for LFQ in PEAKS: peptide quality $\geq 2$; average area $\geq$ 1E5; charge between 2 and 5; used peptides $\geq 1$ and normalization by 10 internal standard proteins.

### Statistical analysis

The resulting protein lists from the LFQ analysis were exported to the Perseus software (*Tyanova et al., 2016*). To reduce the number of random missing values in the dataset, only proteins detected in minimum three replicates in at least one experimental group were kept (*Tyanova & Cox, 2018*). Any missing values were replaced by the average area threshold (1E5) applied in PEAKS before log2 transformation (*Webb-Robertson et al., 2015*; *Lazar et al., 2016*; *Kong et al., 2022*). The overall similarity between the samples were evaluated by principal component analysis (PCA) using the web tool ClustVis (*Metsalu & Vilo, 2015*). Data were centered and scaled by unit variance scaling, and singular value decomposition (SVD) with imputation applied to calculate the principal components. Evaluation of differential protein abundance was performed using one-way analysis of variance (ANOVA) followed by Tukey's HSD post hoc test at 5% FDR in Perseus. Only significantly changed proteins with a fold change $>2$ or $<0.5$ were used for further analysis. Area-proportional Venn diagrams of significantly changed proteins were created using BioVenn (*Hulsen, De Vlieg & Alkema, 2008*).

### Functional enrichment analysis

Goldfish protein sequences of the significantly changed proteins were uploaded to BlastKOALA (*Kanehisa, Sato & Morishima, 2016*) to acquire the Uniprot gene names. The entry names were then matched with the corresponding zebrafish names for enrichment analysis. g:Profiler (version e101_eg48_p14_baf17f0) was used for functional profiling with g:SCS multiple testing correction method and a significance threshold of 0.05 (*Raudvere et al., 2019*). The zebrafish entry list was imported as an ordered query based on fold changes for pairwise comparison of anoxia or reoxygenation to normoxia, and as an unordered query for the entire ANOVA protein list.

## RESULTS

### Tissue-specific adaptations of the proteome in response to anoxia and reoxygenation

To investigate how the proteome of crucian carp changes with oxygen depletion and reoxygenation, we exposed crucian carp to five days normoxia, five days anoxia (<0.1% air saturation) and five days anoxia followed by one day reoxygenation. Extracted proteins from the brain, heart and liver were analyzed by liquid chromatography-mass spectrometry (LC-MS) followed by label-free quantitation (LFQ), resulting in 2722, 2410 and 1997 identified proteins from the brain, heart and liver, respectively.

Normalized and log2 transformed data were subjected to principal component analysis (PCA), revealing that the three investigated tissues responded differently to the three treatments. The brain and liver proteome did not show any specific grouping according to treatment, as demonstrated by the overlap between groups along the PC axes (Fig. 1). The heart was the only tissue to display clustering of the proteome according to oxygen availability. The normoxic group in heart tissue appeared to be most different from the proteome profiles of anoxia and reoxygenation (Fig. 1). Differences in protein levels between the experimental groups were assessed with one-way ANOVA with Tukey's post-hoc test, and only proteins passing Tukey's FDR < 0.05 and fold change > 2 or < 0.5 were considered as changed (Files S1–S3).

The brain was least affected by anoxia and reoxygenation with only 66 proteins significantly changed (Fig. 2). The levels of only 7 proteins increased in abundance during anoxia, and 12 proteins decreased compared to normoxia. In reoxygenation, the abundance of 45 proteins changed, with 33 being increased and 12 proteins decreased compared to normoxia (Fig. 2). The highest number of significantly changed proteins was found in the heart, where 243 proteins changed significantly between the experimental groups (Fig. 2). More specifically, 82 proteins increased and 68 proteins decreased in the heart during anoxia compared to normoxia, whereas 87 proteins increased and 72 decreased in reoxygenation compared to normoxia. In the liver, a total of 135 proteins significantly changed in abundance (Fig. 2). Of these, 87 proteins increased, and five decreased in anoxia compared to normoxia. In reoxygenation, 87 liver proteins increased and nine proteins decreased compared to normoxia.

### Functional enrichment analysis

Significantly changed proteins (FDR < 0.05, fold change > 2 or < 0.5) were uploaded to the g:Profiler software for functional enrichment analysis (also named over-representation analysis). First, all changed proteins (including both increased and decreased protein levels) were analyzed, followed by differentiated analysis in which only increased or decreased proteins were included. Tissue-specific over-representation of mitochondrial proteins was apparent (Fig. 3). While there was no over-representation of mitochondrial proteins in the brain, mitochondrial proteins were over-represented during anoxia-reoxygenation among downregulated proteins in the heart and upregulated proteins in the liver (Fig. 3). When all proteins that changed were included in the functional enrichment analysis, carboxylic acid metabolic process and generation of precursor metabolites and energy were

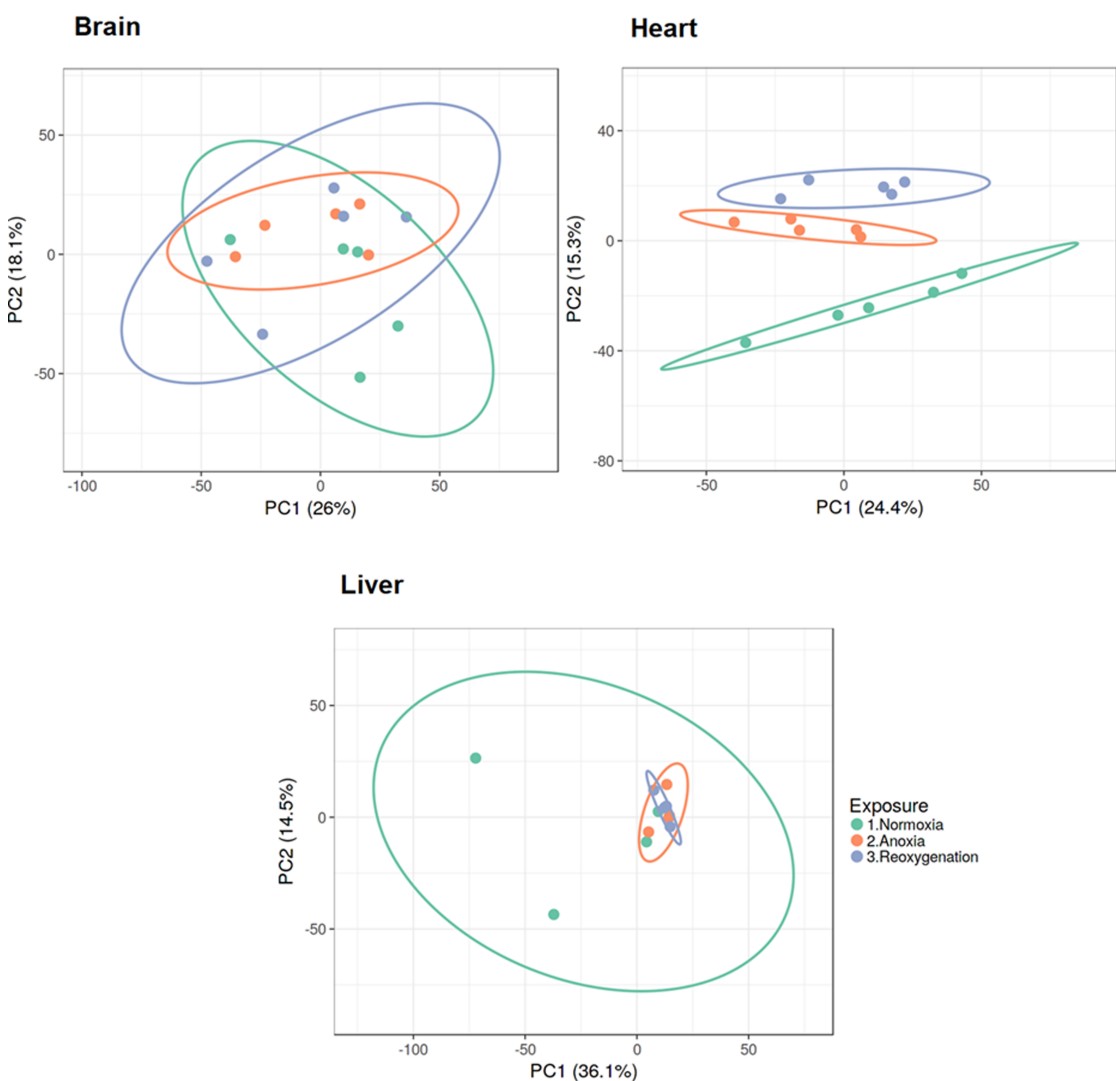

**Figure 1** **PCA plots of brain, heart and liver proteomes during normoxia, anoxia and reoxygenation.** Data were normalized and log2 transformed and uploaded to the ClustVis web tool. Unit variance scaling was applied to rows, and SVD with imputation was used to calculate the principal components. Eclipses illustrate 95% confidence intervals.

over-represented terms in both the heart and the liver (Fig. 3). Aerobic respiration and proteolysis were over-represented terms among all changed proteins in the heart, and in the liver amino acid degradation (valine, leucine, isoleucine) pathway and oxidoreductase activity were over-represented among all regulated proteins (Fig. 3). In the brain, proteins involved in aminoacyl-tRNA biosynthesis were over-represented during reoxygenation (Fig. 3). Proteolysis, including complement activation was over-represented among proteins that increased in amount during anoxia in the heart, and finally, succinate-CoA metabolic process was over-represented among increased proteins during anoxia and reoxygenation in the liver (Fig. 3).

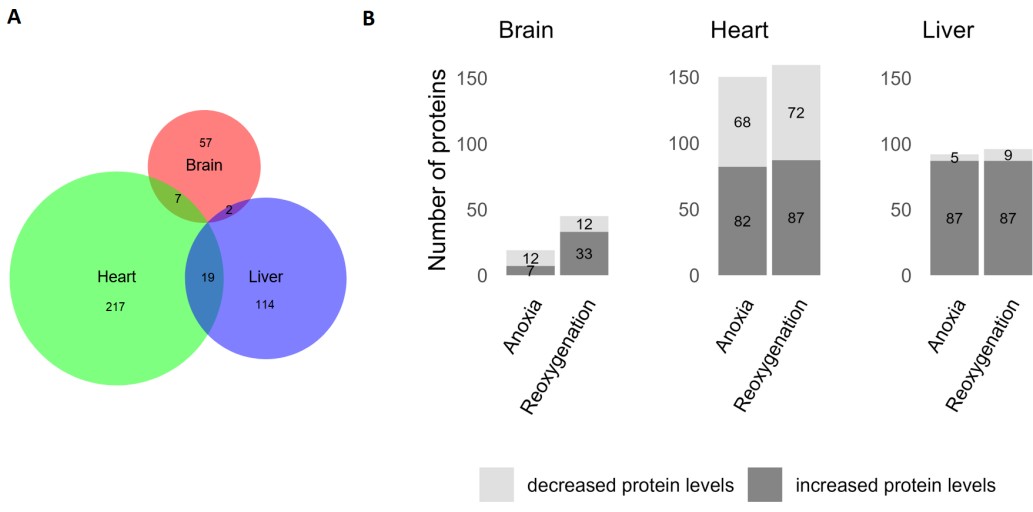

**Figure 2** **Significantly changed proteins in the brain, heart and liver during anoxia-reoxygenation compared to normoxic control.** (A) Number of changed proteins in each tissue. Size corresponds to number of proteins. (B) Number of increased and decreased proteins in anoxia and reoxygenation compared to normoxia in the brain, heart and liver.

## Regulated pathways

Mitochondrial protein levels changed in both heart and liver, and not only related to the ETS, but also to the mitochondrial membrane integrity. Furthermore, since the redistribution of energy sources is crucial to survive long-term anoxia, we were particularly interested in proteins involved in glucose metabolism and oxidative phosphorylation, and we examined whether protein levels of ETS proteins were altered in response to anoxia. In addition, proteins predicted to be involved in the complement system were over-represented in the anoxic heart. Thus, we set out to analyze these pathways in detail, covered in the following sections.

## Mitochondrial transport and structure

The functional enrichment analysis revealed an over-representation of the mitochondrion cellular compartment (Fig. 3). A closer examination of the over-represented proteins revealed that the protein levels of mitochondrial membrane transporters and proteins involved in maintaining mitochondrial membrane integrity were altered during anoxia-reoxygenation in the heart and the liver (Files S2 & S3). In the heart, several of the proteins involved in protein import decreased in protein levels (Fig. 4 & Table 1). Among them, mitochondrial import receptor subunit TOM40 (TOMM40) was only detected in the normoxic control group, mitochondrial inner membrane translocase subunit Tim9 (TIMM9) decreased (0.33 fold change) in anoxia while the Tim10 subunit (TIMM10) decreased (0.46 fold change) in reoxygenation. Also situated in the outer mitochondrial membrane and part of the mitochondrial permeability transition pore (PTP), voltage-dependent anion-selective channel protein 1 (VDAC) decreased (0.37 fold change) in the anoxic heart (Fig. 4 & Table 1). In the reoxygenated heart, the mitochondrial contact site and cristae junction organizing system (MICOS) complex components MIC26 and MIC19

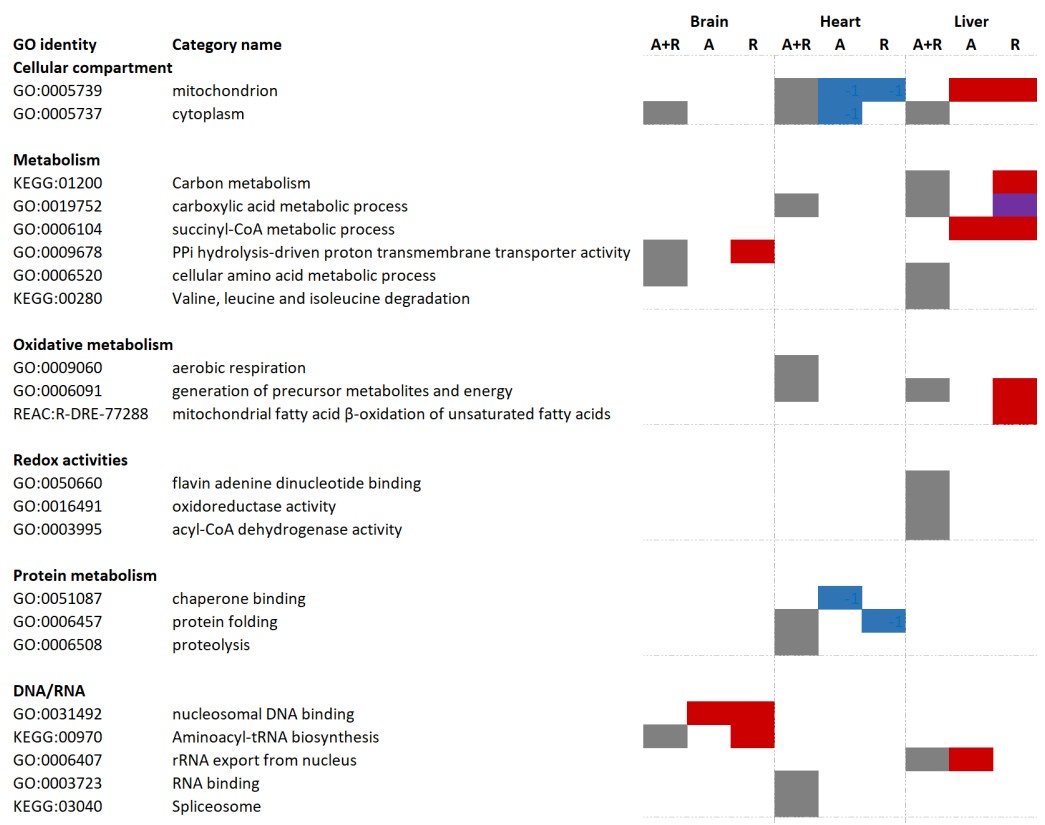

|  |  | Brain | | | Heart | | | Liver | | |
|---|---|---|---|---|---|---|---|---|---|---|
| GO identity | Category name | A+R | A | R | A+R | A | R | A+R | A | R |
| **Cellular compartment** | | | | | | | | | | |
| GO:0005739 | mitochondrion | | | | | | | | | |
| GO:0005737 | cytoplasm | | | | | | | | | |
| **Metabolism** | | | | | | | | | | |
| KEGG:01200 | Carbon metabolism | | | | | | | | | |
| GO:0019752 | carboxylic acid metabolic process | | | | | | | | | |
| GO:0006104 | succinyl-CoA metabolic process | | | | | | | | | |
| GO:0009678 | PPi hydrolysis-driven proton transmembrane transporter activity | | | | | | | | | |
| GO:0006520 | cellular amino acid metabolic process | | | | | | | | | |
| KEGG:00280 | Valine, leucine and isoleucine degradation | | | | | | | | | |
| **Oxidative metabolism** | | | | | | | | | | |
| GO:0009060 | aerobic respiration | | | | | | | | | |
| GO:0006091 | generation of precursor metabolites and energy | | | | | | | | | |
| REAC:R-DRE-77288 | mitochondrial fatty acid β-oxidation of unsaturated fatty acids | | | | | | | | | |
| **Redox activities** | | | | | | | | | | |
| GO:0050660 | flavin adenine dinucleotide binding | | | | | | | | | |
| GO:0016491 | oxidoreductase activity | | | | | | | | | |
| GO:0003995 | acyl-CoA dehydrogenase activity | | | | | | | | | |
| **Protein metabolism** | | | | | | | | | | |
| GO:0051087 | chaperone binding | | | | | | | | | |
| GO:0006457 | protein folding | | | | | | | | | |
| GO:0006508 | proteolysis | | | | | | | | | |
| **DNA/RNA** | | | | | | | | | | |
| GO:0031492 | nucleosomal DNA binding | | | | | | | | | |
| KEGG:00970 | Aminoacyl-tRNA biosynthesis | | | | | | | | | |
| GO:0006407 | rRNA export from nucleus | | | | | | | | | |
| GO:0003723 | RNA binding | | | | | | | | | |
| KEGG:03040 | Spliceosome | | | | | | | | | |

**Figure 3** **Functional enrichment analyses of regulated brain, heart and liver proteins during anoxia and reoxygenation.** Grey boxes: over-represented category when all changed proteins (including both increased and decreased proteins) were included. Red boxes display over-representation among increased proteins compared to normoxia; blue boxes display over-representation among decreased proteins compared to normoxia; purple box display over-representation among both increased and decreased proteins compared to normoxic control. Selection of over-represented pathways of interest (FDR 5%). A, anoxia; R, reoxygenation; A + R, anoxia and reoxygenation combined, GO, Gene Ontology; KEGG, Kyoto Encyclopedia of Genes and Genomes; REAC, Reactome pathway database. Grouping of GO identities are not linked to any formal GO classification.

decreased (0.19 and 0.44 fold change, respectively). Increased protein levels in the heart were apparent for sorting and assembly machinery component 50 (SAMM50), increasing 13.21-fold in anoxia and 5.89-fold in reoxygenation, and the outer membrane translocase TOM1-like protein 2 (TOM1L2) that only reached detectable levels during reoxygenation (Fig. 4 & Table 1).

In contrast to the decrease of cardiac TIM subunits in anoxia and reoxygenation, TIM8 increased 18.13-fold in the reoxygenated liver (not being detected in normoxic control) (Fig. 5 & Table 1). The mitochondrial uncoupling protein (UCP2) was also detected primarily during reoxygenation in the liver, leading to a 12.48-fold increase compared to normoxia. Also in the liver, mitochondrial fission protein (FIS1) increased 12.53-fold in anoxia and 8.43-fold in reoxygenation, while VDAC2 increased 2.70-fold

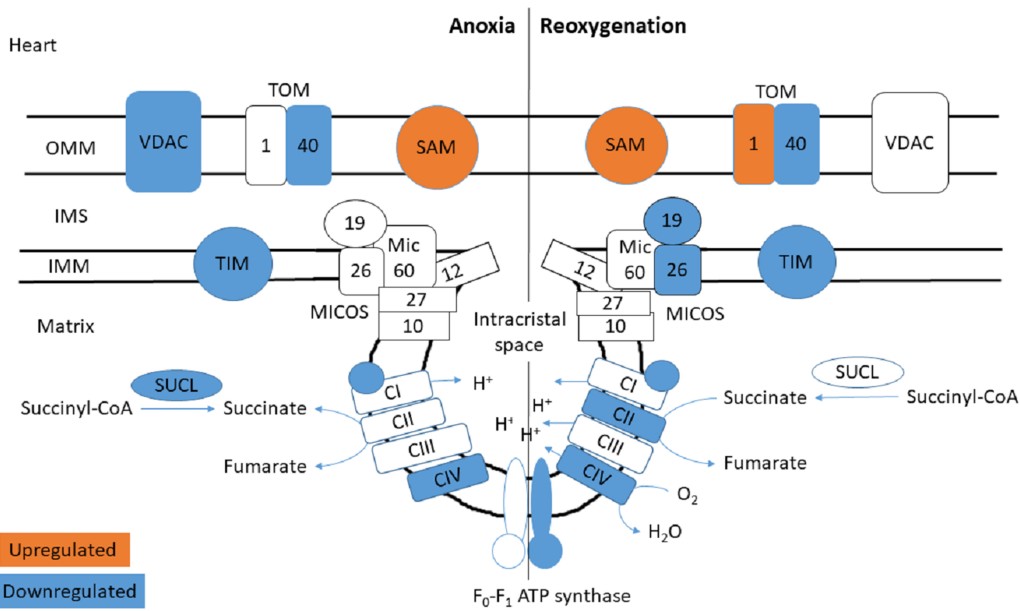

**Figure 4  Mitochondrial membrane proteins in the heart during anoxia and reoxygenation.** Significantly changed proteins as compared to normoxia are depicted in orange (upregulated) and blue (downregulated). Details are found in Table 1. Schematic presentation not showing all proteins related to cristae forming. SUCL, Succinate-CoA ligase; C1-CIV, ETC complexes I-IV; TIM, mitochondrial import inner membrane translocase; MICOS, mitochondrial contact site and cristae junction organizing system; VDAC, voltage-dependent anion-selective channel protein; TOM, translocase of the outer membrane; SAM, sorting and assembly machinery component. OMM, outer mitochondrial membrane; IMS, intermembrane space; IMM, inner mitochondrial membrane.

during reoxygenation (Fig. 5 & Table 1). Of mitochondrial proteins in brain, only the SAM-complex component metaxin-2 (MTX2) was altered, increasing 9-fold in reoxygenation (File S1).

## Aerobic respiration: TCA cycle and oxidative phosphorylation

During anoxia, oxidation of acetyl-CoA in the TCA cycle and oxidative phosphorylation are stalled due to the lack of oxygen as final electron acceptor for the ETS. Concordantly, proteins predicted to be involved in aerobic respiration were altered in the heart (Fig. 3). In the heart, both cytoplasmic (IDH1) and mitochondrial (IDH3B) isocitrate dehydrogenases decreased during anoxia-reoxygenation compared to normoxia (0.47 and 0.25 fold change, respectively; Table 2), while succinate dehydrogenase flavoprotein subunit (SDHA; Complex II of the ETS) decreased in reoxygenation compared to normoxia (0.08 fold change; Fig. 4 & Table 2). Succinate-CoA ligase was one of the few proteins that changed in more than one tissue, showing tissue-specific changes with decreasing levels in the heart (SUCLG2; 0.04 fold change) and increasing levels in the liver (SUCLA; 7.26-fold and SUCLG2; 5.95-fold) during anoxia (Figs. 5–6 & Table 2). After one day of reoxygenation, cardiac succinate-CoA ligase protein levels had returned to normoxic values (Figs. 5–6 & Table 2) while in liver the protein levels continued to increase (7–14-fold).

**Table 1  Regulated proteins involved in mitochondrial transport and structure in the heart and the liver.**

| Protein name | Gene name | Accession number | FDR -adjusted $p$-value | Ratio | |
|---|---|---|---|---|---|
| | | | | A $vs$ N | R $vs$ N |
| *Heart* | | | | | |
| MICOS complex subunit MIC26 isoform X1 | APOOA | XP_026096387.1 | 0.0017 | **0.69** | **0.19** |
| MICOS complex subunit Mic19-like isoform X1 | CHCHD3A | XP_026085708.1 | 0.0018 | 0.85 | **0.44** |
| Sorting and assembly machinery component 50 homolog B-like | SAMM50 | XP_026065790.1 | 0.0004 | **13.21** | **5.89** |
| Mitochondrial import inner membrane translocase subunit Tim10-like | TIMM10 | XP_026101102.1 | <0.0001 | 0.81 | **0.46** |
| Mitochondrial import inner membrane translocase subunit Tim9 | TIMM9 | XP_026142623.1 | 0.0055 | **0.33** | 1.23 |
| TOM1-like protein 2 isoform X4 | TOM1L2 | XP_026078644.1 | 0.0001 | 8.54 | **43.94** |
| Voltage-dependent anion-selective channel protein 1 | VDAC1 | XP_026089990.1 | 0.0023 | **0.37** | 0.61 |
| *Liver* | | | | | |
| Mitochondrial fission 1 protein-like | FIS1 | XP_026122626.1 | 0.0034 | **12.53** | **8.43** |
| Mitochondrial import inner membrane translocase subunit Tim8 A-like | TIMM8A | XP_026103899.1 | 0.0003 | 2.88 | **18.13** |
| Mitochondrial uncoupling protein 2-like | UCP2 | XP_026085469.1 | 0.0476 | 6.61 | **12.48** |
| Voltage-dependent anion-selective channel protein 2 | VDAC2 | XP_026080095.1 | 0.0361 | 2.45 | **2.70** |

**Notes.**

Ratios between the experimental groups shown in bold satisfy the statistical threshold (one-way ANOVA followed by Tukey's post-hoc test with FDR < 0.05 and fold change >2 or <0.5) compared to normoxic control.

A, anoxia; N, normoxia; R, reoxygenation.

Ratios in Roman: FDR > 0.05.

Similar to decreasing TCA protein levels, several ETS proteins decreased in the heart during anoxia-reoxygenation (Fig. 4 & Table 2). Protein levels of NADH dehydrogenase [ubiquinone] 1 beta subcomplex subunit 10 (NDUFB2), an accessory subunit of Complex I, decreased in the heart during anoxia (0.26 fold change) and reoxygenation (0.08 fold change) compared to normoxia. The cardiac protein levels of cytochrome C oxidase (COX; Complex IV) subunits decreased in anoxia (COX4I1; 0.28 fold change) and reoxygenation (COX5B1; 0.34 fold change and COX7C; 0.16 fold change) compared to normoxic control. During reoxygenation, the protein levels of ATP synthase subunit O (Complex V; ATP5PO), part of the peripheral stalk and $F_o$ domain, decreased (0.50-fold) in the heart compared to normoxia (Fig. 4 & Table 2).

In contrast to decreased protein levels in heart, TCA and ETS related proteins increased in the liver during anoxia and reoxygenation (Fig. 5 & Table 2). The Complex I accessory subunit NADH dehydrogenase [ubiquinone] 1 beta subcomplex subunit 3 (NDUFB3), increased during both anoxia (13.97-fold) and reoxygenation (22.74-fold) in the liver (Fig. 5 & Table 2), while the Complex I core subunit NADH-ubiquinone oxidoreductase 75 kDa subunit (NDUFS1) increased significantly in anoxia (3.81-fold), and not significantly during reoxygenation (Table 2), all compared to normoxia. The protein levels of Complex IV subunit 5B (COX5B2) increased 7.71-fold in anoxia and 5.81-fold in reoxygenation, while the Complex IV subunit 6B1 (COX6B1) increased 2.62-fold in reoxygenation compared to normoxic control (Fig. 5 & Table 2). No significant changes in protein levels of TCA or ETS related proteins were observed in the brain.

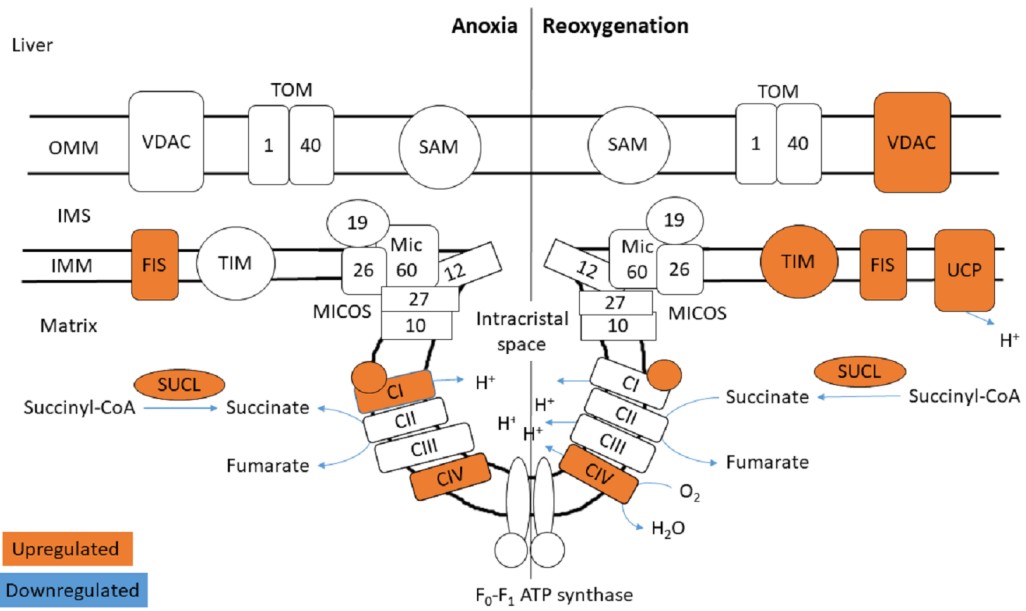

**Figure 5** **Mitochondrial membrane proteins in the liver during anoxia and reoxygenation.** Significantly changed proteins as compared to normoxia are depicted in orange (upregulated) and blue (downregulated). Details are found in Table 1. Schematic presentation not showing all proteins related to cristae forming. SUCL, Succinate-CoA ligase; C1-CIV, ETC complexes I-IV; TIM, mitochondrial import inner membrane translocase; MICOS, mitochondrial contact site and cristae junction organizing system; VDAC, voltage-dependent anion-selective channel protein; TOM, translocase of the outer membrane; SAM, sorting and assembly machinery component; UCP, mitochondrial uncoupling protein; FIS, mitochondrial fission 1 protein. OMM, outer mitochondrial membrane; IMS, intermembrane space; IMM, inner mitochondrial membrane.

## Glycogen metabolism

Glycogen metabolism is fundamental for anoxic crucian carp survival, and considerable change in the abundance of relevant proteins was observed in the heart and liver. Both glycogen synthesizing and degrading protein levels changed during anoxia-reoxygenation. In the heart, glycogen debranching enzyme (AGLA) increased 10.1-fold and 8.2-fold during anoxia and reoxygenation, respectively, compared to normoxia (Fig. 6A & Table 3). Furthermore, glycogen phosphorylase (PYGB) increased (1.7-fold) in anoxia compared to normoxia, and the key glycolytic regulator hexokinase-2 (HK2) increased in both anoxia and reoxygenation (2.4-fold) compared to normoxia. Among glycogen synthesizing proteins, glycogenin (GYG1A) increased in anoxia (2.8-fold) and reoxygenation (1.8-fold), and glycogen synthase (GYS1) increased in anoxia (2.2-fold) in the heart, all compared to normoxia (Fig. 6B & Table 3). In the liver, only phosphorylase b kinase (PHKB) increased during anoxia-reoxygenation (3.4 to 4.6-fold) compared to normoxia, while glycogen debranching enzyme (AGLA) decreased slightly (0.7-fold change) in reoxygenation compared to normoxia (Fig. 6B & Table 3). In the brain, glycogen synthase was the only protein that changed during anoxia and reoxygenation, increasing 3-fold in both anoxia and reoxygenation compared to normoxia (File S1).

**Table 2  Regulated proteins involved aerobic respiration in the heart and the liver.**

| Protein name | Gene name | Accession number | FDR-adjusted *p*-value | Ratio | |
|---|---|---|---|---|---|
| | | | | A *vs* N | R *vs* N |
| *Heart* | | | | | |
| Isocitrate dehydrogenase [NADP] cytoplasmic-like | IDH1 | XP_026073874.1 | 0.0001 | **0.47** | **0.51** |
| Isocitrate dehydrogenase [NAD] subunit beta mitochondrial-like isoform X1 | IDH3B | XP_026052197.1 | 0.0124 | **0.25** | 0.64 |
| Succinate–CoA ligase [GDP-forming] subunit beta mitochondrial-like | SUCLG2 | XP_026131177.1 | 0.0177 | **0.04** | 0.81 |
| NADH dehydrogenase [ubiquinone] 1 beta subcomplex subunit 2 mitochondrial-like | NDUFB2 | XP_026087107.1 | <0.0001 | **0.26** | **0.08** |
| Succinate dehydrogenase [ubiquinone] flavoprotein subunit mitochondrial-like | SDHA | XP_026145409.1 | 0.0278 | 0.22 | **0.08** |
| Cytochrome c oxidase subunit 4 isoform 1 mitochondrial | COX4I1 | XP_026087014.1 | 0.0000 | **0.28** | 0.83 |
| Cytochrome c oxidase subunit 5B mitochondrial-like | COX5B1 | XP_026130115.1 | 0.0390 | 0.61 | **0.34** |
| Cytochrome c oxidase subunit 7C mitochondrial-like | COX7C | XP_026066873.1 | 0.0199 | 1.03 | **0.16** |
| ATP synthase subunit O mitochondrial-like | ATP5PO | XP_026073676.1 | 0.0078 | 0.76 | **0.50** |
| *Liver* | | | | | |
| Succinate–CoA ligase [ADP-forming] subunit beta mitochondrial | SUCLA2 | XP_026070025.1 | <0.0001 | **7.26** | **14.39** |
| Succinate–CoA ligase [GDP-forming] subunit beta mitochondrial-like | SUCLG2 | XP_026131177.1 | 0.0078 | **5.95** | **7.14** |
| NADH dehydrogenase [ubiquinone] 1 beta subcomplex subunit 3-like isoform X3 | NDUFB3 | XP_026128397.1 | <0.0001 | **13.97** | **22.74** |
| NADH-ubiquinone oxidoreductase 75 kDa subunit mitochondrial-like | NDUFS1 | XP_026115071.1 | 0.0353 | **3.81** | 2.93 |
| Cytochrome c oxidase subunit 5B mitochondrial-like | COX5B2 | XP_026133598.1 | 0.0096 | **7.71** | **5.81** |
| Cytochrome c oxidase subunit 6B1-like isoform X2 | COX6B1 | XP_026114188.1 | 0.0009 | 1.51 | **2.62** |

**Notes.**
Ratios between the experimental groups shown in bold satisfy the statistical threshold (one-way ANOVA followed by Tukey's post-hoc test with FDR < 0.05 and fold change >2 or <0.5) compared to normoxic control.
A, anoxia; N, normoxia; R, reoxygenation.
Ratios in Roman: FDR > 0.05.

## Complement system

The complement system is comprised of several soluble and membrane-bound proteins and is a key part of the immune system response (*Reis et al., 2019*). In the heart, proteins predicted to be involved in proteolysis and more specifically the activation of the complement system were over-represented during anoxia (Fig. 3 & File S2). More specifically, the complement components C3a.1, C4, C5 and C7 and complement factor H increased 3–5-fold in anoxia compared to normoxia (Table 4). In addition, the coagulation related factors prothrombin and coagulation factor X increased 2-fold and 6-fold in anoxia compared to normoxia, respectively (Table 4). Our data also suggest that the complement system is maintained in an active state during reoxygenation since the complement factor I, which degrades C3 and C4 (*Ricklin et al., 2010*), decreased in reoxygenation (0.44 fold change) compared to normoxia. In addition, complement decay accelerating factor, which

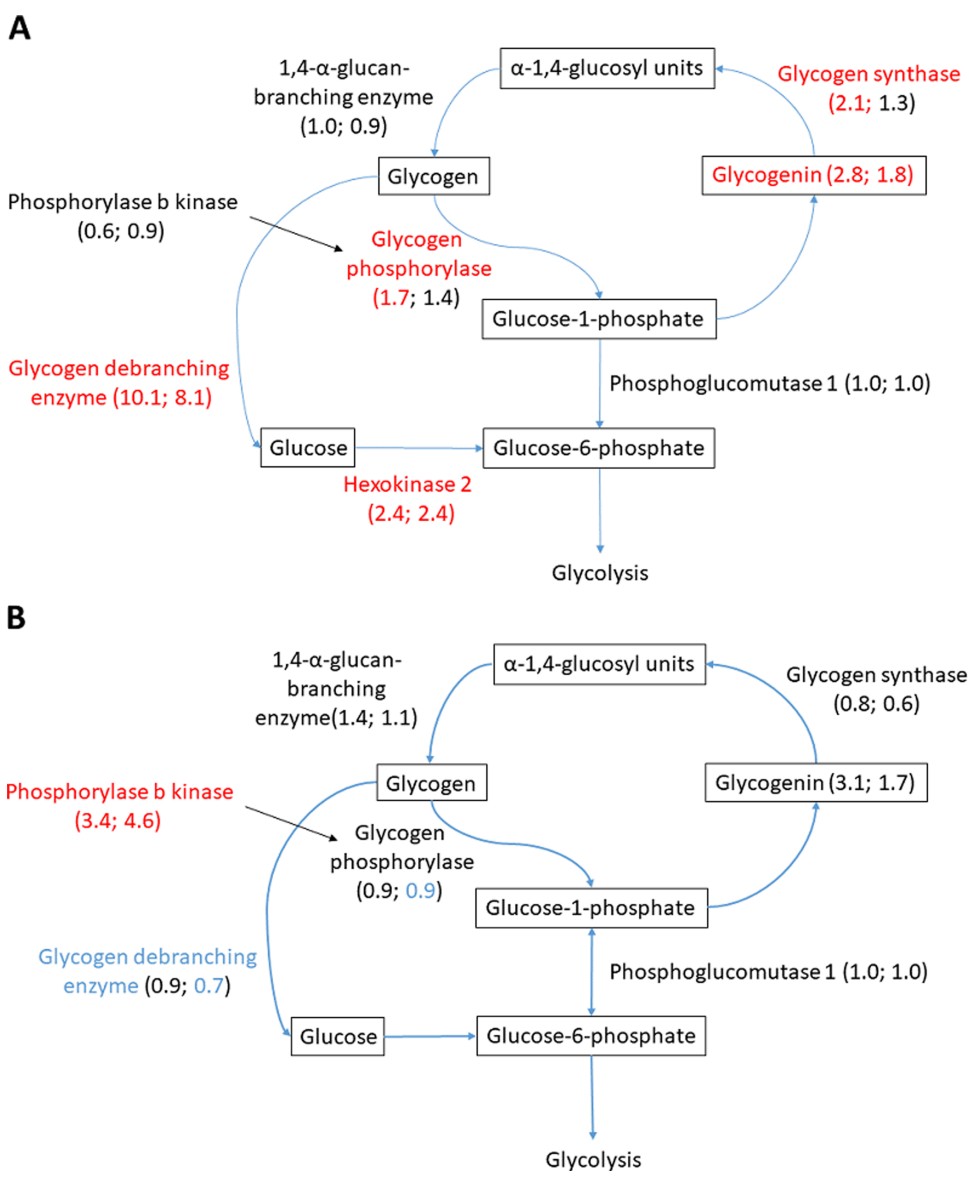

**Figure 6 Glycogen metabolism in (A) heart and (B) liver.** Changed proteins meeting the statistical requirements of FDR-adjusted *p*-value < 0.05 are colored in red (increased compared to normoxia) and blue (decreased compared to normoxia). Fold change in parenthesis, first number showing fold change A/N, and second number R/N. Changes not passing statistical test at FDR < 0.05 are colored in black.

inhibits C3 activation (*Sun et al., 1999*), decreased in anoxia (0.14 fold change) and in reoxygenation (0.37 fold change) compared to normoxia (Table 4).

## DISCUSSION

### Global proteome responses to anoxia-reoxygenation

The PCA analysis and hierarchical clustering revealed that the three investigated tissues responded differently to anoxia and reoxygenation. The brain proteome did not show

**Table 3** **Proteins involved in glycogen metabolism in the heart and the liver.**

| Protein name | Gene name | Accession | FDR-adjusted *p*-value | Ratio | |
| --- | --- | --- | --- | --- | --- |
| | | | | A/N | R/N |
| *Heart* | | | | | |
| Glycogen debranching enzyme-like isoform X1 | AGLA | XP_026062230.1 | 0.0012 | **10.1** | **8.2** |
| Glycogen phosphorylase muscle form-like | PYGB | XP_026122785.1 | 0.0026 | **1.7** | 1.4 |
| Hexokinase-2-like | HK2 | XP_026066335.1 | <0.0001 | **2.4** | **2.4** |
| Glycogenin-1-like | GYG1A | XP_026062454.1 | <0.0001 | **2.8** | **1.8** |
| Glycogen [starch] synthase muscle-like | GYS1 | XP_026063840.1 | 0.0005 | **2.2** | 1.3 |
| 1,4-alpha-glucan-branching enzyme-like | GBE1 | XP_026138510.1 | 0.8553 | 1.0 | 0.9 |
| Phosphorylase b kinase regulatory subunit alpha | PHKB | XP_026130815.1 | 0.2528 | 0.6 | 0.9 |
| Phosphoglucomutase-1-like | PGM1 | XP_026069410.1 | 0.9737 | 1.0 | 1.0 |
| *Liver* | | | | | |
| Phosphorylase b kinase regulatory subunit beta | PHKB | XP_026071436.1 | 0.0087 | **3.4** | **4.6** |
| Glycogen debranching enzyme-like isoform X1 | AGLA | XP_026062230.1 | 0.0164 | 0.9 | **0.7** |
| Glycogen phosphorylase liver form | PYGB | XP_026079461.1 | 0.0248 | 0.9 | **0.9** |
| Glycogenin-1-like | GYG1A | XP_026062454.1 | 0.1200 | 1.3 | 1.3 |
| Glycogen [starch] synthase liver-like | GYS1 | XP_026089611.1 | 0.5011 | 0.8 | 0.6 |
| 1,4-alpha-glucan-branching enzyme-like | GBE1 | XP_026138510.1 | 0.2276 | 1.4 | 1.1 |
| Phosphoglucomutase-1-like | PGM1 | XP_026069410.1 | 0.9500 | 1.0 | 1.0 |

**Notes.**
Ratios between the experimental groups shown in bold satisfy the statistical threshold (one-way ANOVA followed by Tukey's post-hoc test with FDR < 0.05 and fold change >2 or <0.5) compared to normoxic control.
A, anoxia; N, normoxia; R, reoxygenation.
Ratios in Roman: FDR > 0.05.

any specific grouping according to oxygen exposure, demonstrated by complete overlap between the groups along the component axes (Fig. 1). This is in line with the described unaffected or possibly slightly suppressed protein synthesis rates (*Smith et al., 1996*) and modest changes in protein levels in crucian carp brain during anoxia (*Smith et al., 2009*). Still, this does not mean that the significant changes seen in some brain protein levels are not of consequence for anoxic survival. Because of the key role of the liver in anoxic survival and the heavily repressed protein synthesis during anoxia (*Smith et al., 1996*), we expected the liver proteome to cluster according to the exposure groups. However, no global proteome response was apparent (Fig. 1). Despite the variation within the normoxic group, the anoxic and reoxygenated groups seem to be overlapping as observed in the brain. The heart proteome was the only tissue that clustered according to the experimental groups, suggesting that the cardiac proteome is altered to a higher degree than the brain and liver proteomes in response to oxygen variations.

For heart tissue the reoxygenated group was closer to the anoxic group than the normoxic one. It is possible that more than 24 h reoxygenation is needed for the proteome to return to the normoxic situation. Indeed, cardiac transcript regulation in response to oxygen depletion had not reached its maximum after one day anoxia (*Stensløkken et al., 2014*). Since overall protein synthesis in crucian carp is a rather slow process (*Smith et al., 1996*), it seems plausible that changes in the majority of protein levels are not part of an instant

**Table 4  Regulated cardiac proteins involved in the complement system.**

| Protein name | Gene name | Accession | FDR-adjusted *p*-value | Ratio | |
|---|---|---|---|---|---|
| | | | | A/N | R/N |
| Complement C3-like | C3A.1 | XP_026068498.1 | 0.0293 | **3.48** | 3.04 |
| Complement C3-like | C3A.2 | XP_026068420.1 | 0.0324 | 5.18 | **14.84** |
| Complement C4-like | C4 | XP_026082535.1 | 0.0004 | **3.62** | **3.64** |
| Complement C5-like | C5 | XP_026108101.1 | 0.0008 | **2.69** | **3.21** |
| Complement component C7-like | C7A | XP_026117557.1 | 0.0097 | **4.56** | 3.38 |
| Complement decay-accelerating factor-like isoform X1 | – | XP_026118213.1 | 0.0092 | **0.14** | **0.37** |
| Complement factor H-like | CFHL2 | XP_026091802.1 | 0.0008 | **2.98** | 1.55 |
| Complement factor H-like | CFHL3 | XP_026092117.1 | 0.0115 | **2.00** | 1.26 |
| Complement factor H-like isoform X10 | CFH | XP_026091793.1 | 0.0040 | **2.88** | 1.79 |
| Complement factor I-like | CFI | XP_026093116.1 | 0.0002 | 0.68 | **0.44** |
| Prothrombin-like | F2 | XP_026123457.1 | 0.0260 | **2.00** | **2.07** |
| Coagulation factor X-like | F10 | XP_026111350.1 | 0.0187 | **6.22** | 3.58 |

**Notes.**
Ratios between the experimental groups shown in bold satisfy the statistical threshold (one-way ANOVA followed by Tukey's post-hoc test with FDR < 0.05 and fold change >2 or <0.5) compared to normoxic control.

A, anoxia; N, normoxia; R, reoxygenation.

Ratios in Roman: FDR > 0.05.

response to oxygen depletion or return. Additionally, protein degradation is an ATP-driven process, so it might not be energetically favorable to spend ATP on degrading proteins synthesized during anoxia already after one day reoxygenation. The latter scenario seems plausible especially in the liver, but also in the heart (Fig. 1). Collectively, other means of regulating protein activity are likely to be responsible for the majority of responses to oxygen depletion and return, such as substrate availability, allosteric factors and post-translational modifications.

Nevertheless, the abundance of 66 proteins in brain, 243 proteins in heart and 135 proteins in liver were significantly changed and were subjected to a functional enrichment analysis. Few common responses between the three examined tissues were apparent, indicating that they share few global proteomic responses to cope with the oxygen variations, and that different processes are prioritized in the tissues. The fact that not even oxygen-related terms such as aerobic respiration were over-represented in all tissues, emphasizes that regulation of protein levels is unlikely to be a main regulatory mechanism during anoxia. However, the data do suggest that regulation of protein levels in the mitochondria plays a role during anoxia and reoxygenation, since mitochondria-related terms were over-represented in both the heart and the liver.

## TCA cycle and oxidative phosphorylation

Proteins involved in the TCA-cycle and ETS were altered in a tissue-specific manner during anoxia and reoxygenation. Since anaerobic glycolysis is the main energy source during periods of oxygen limitation regardless of tissue, one may expect to find a decrease (or no change) of aerobic processes in some tissues. Indeed, the abundance of enzymes involved in TCA or ETS did not change in the brain and decreased in the heart (Fig. 4 & Table 2).

In contrast, the proteins that changed in the liver tissue increased in abundance (Fig. 5 & Table 2).

We propose that the increased levels of ETS proteins in the liver are linked to a special role for this organ in handling succinate accumulated during anoxia and metabolized during reoxygenation. As mentioned, succinate accumulated in several tissues in the crucian carp during anoxia, and by far most in the liver (*Dahl et al., 2021*). The proteomic data presented in the current study suggest that succinate handling is tissue specific. Succinate dehydrogenase (Complex II of ETS) decreased during reoxygenation in the heart (Fig. 4 & Table 2), while no changes of this complex were apparent in the brain or the liver. Furthermore, succinate-CoA ligase protein levels were tissue-specifically altered with decreasing protein levels in the heart during anoxia (Fig. 4 & Table 2), a massive (6–14-fold) increase in the liver (Fig. 5 & Table 2) during anoxia and reoxygenation, and no changes in brain. Thus, the route of succinate generation from succinate-CoA is more likely to occur in liver tissue than in the heart of crucian carp.

Of particular importance may be the elevated protein levels of Complex I and IV in the anoxic liver (Fig. 5 and Table 2). This is especially striking since overall hepatic protein synthesis decreases by >95% in anoxia (*Smith et al., 1996*). Together with the massive increase in the uncoupling protein UCP2 (Fig. 5 & Table 1) these changes could make the liver mitochondria especially well-suited to handle high succinate levels during reoxygenation without significant ROS production. Rapid succinate metabolism would lead to a high rate of $H^+$ pumping by the ETS, but if this is coupled with an increased influx of $H^+$ by the UCP2 and the ATP synthase (Complex V), the mitochondria could have sufficient capacity to handle the electrons generated without any electron leak (and therefore ROS generation). Indeed, the ROS production by Complex I has been shown to be greatly reduced by a reduction of the $H^+$ gradient over the mitochondrial inner membrane (*Lambert & Brand, 2004*). A key difference from the mammalian (anoxia-intolerant) situation is that anoxic crucian carp tissues accumulate ADP during anoxia (for example, being more than doubled in the liver (*Dahl et al., 2021*)), while ADP is depleted in anoxia-intolerant animals. Having high ADP levels during reoxygenation will allow Complex V to effectively harvest a high rate of $H^+$ pumping by the ETS and at the same time produce ATP. Indeed, we have previously found that liver ATP levels were significantly higher after 3 and 24 h of reoxygenation than during the normoxic period preceding anoxia (*Dahl et al., 2021*).

UCP2 uncoupling of mitochondrial respiration and ATP production has been shown to reduce ROS generation from ETS complexes (*Cadenas, 2018*; *Tian et al., 2018*; *Zhao et al., 2019*). In a typical scenario of reoxygenation of mammalian (anoxia-intolerant) cells, with excess ETS substrate (NADH for complex I and $FADH_2$ for Complex II), the high proton concentration gradient can stimulate RET-induced ROS (*Chouchani et al., 2016*) and lead to slow electron transport and partial reduction of molecular oxygen, forming superoxide (*Baffy, 2017*). However, enhanced UCP2-induced proton leak into the mitochondrial matrix during reoxygenation would relieve the increasing proton gradient and thus diminish the ROS production. Notably, UCP2-deficient mice have been found to show higher levels of ROS (*Arsenijevic et al., 2000*).
The heart may use a different strategy to minimize ROS production during reoxygenation. Our metabolomic study (*Dahl et al., 2021*) suggested that the energetic recovery of the heart during reoxygenation is slowed down compared to other tissues, which could be aided by the fact that the crucian carp heart primarily receives oxygen poor blood returning from the systemic circulation. Moreover, succinate export to the liver as well as reduced proteins levels of succinate producing enzymes (succinate dehydrogenase and succinate-CoA ligase) in the anoxic heart could aid in repressing the rate of succinate accumulation. Indeed, Complex II protein levels also decreased in the heart of anoxia-tolerant western painted turtles (*Chrysemys picta bellii*) during cold acclimatization, which was suggested to have a function in reducing succinate accumulation during anoxia (*Alderman et al., 2021*). We found reduced protein levels of an accessory subunit of Complex I in the anoxic and reoxygenated heart (Table 2), and it may be that RET can be hindered by reduced Complex I activity (*Chouchani et al., 2016*; *Blaza, Vinothkumar & Hirst, 2018*), especially if no other components involved in electron transport are increased (as in the liver). Thus, in the heart the decrease of cardiac ETS complexes could have a protecting role against ROS damage during reoxygenation.

Decreased transcription and activity of the ATP synthase (Complex V) have been seen in anoxic crucian carp heart (*Stensløkken et al., 2014*). In this present study, there was a non-significant indication of a decrease in ATP synthase protein levels in the heart during anoxia, and a significant decrease during reoxygenation (Table 2). A decrease in ATP synthase levels during anoxia could suppress the amount of ATP consumed for maintaining mitochondrial membrane potential (*Galli, Lau & Richards, 2013*; *Stensløkken et al., 2014*). In reoxygenation, however, reduced ATP synthase protein levels could be counterproductive, but is in line with the observed slower energetic recovery of the crucian carp heart after anoxia (*Dahl et al., 2021*). Combined with decreased protein levels of VDAC1 (Table 2), part of the mitochondrial permeability transition pore assembly (MPTP) (*Karch & Molkentin, 2014*), we speculate that a decrease in abundance of these proteins could be implicated in regulation of the MPTP. MPTP opening is triggered by ROS, leading to depolarization of the mitochondrial membrane and eventually cell death (*Baines, 2009*). Thus, decreased protein levels of its components during reoxygenation, when ROS production could increase, would contribute to mitochondrial protection.

## Mitochondrial transport and structure

Apart from the ETS enzymes, we found changes in abundance of mitochondrial transport protein and proteins involved in maintaining the integrity of mitochondrial membranes and the cristae junctions. The mitochondrial contact site and cristae junction organizing system (MICOS) has an essential role in maintaining the cristae junctions of the inner membrane (*Kozjak-Pavlovic, 2017*). Together with the sorting and assembly machinery (SAM) complex, MICOS has been proposed to indirectly regulate respiratory rates by cristae junction formation to facilitate ETS supercomplex assemblies (*Cogliati, Enriquez & Scorrano, 2016*). The supercomplex formation of the ETS complexes in folded cristae has been suggested to result in a proton concentration gradient from Complex I that accumulates at Complex V at the concave tip of the cristae junction (*Davies et al., 2011*).

In the heart of reoxygenated crucian carp, two MICOS complex components (MIC26 and MIC19) decreased in protein levels, while MIC60 (Mitofilin) protein levels decreased non-significantly (0.5-fold change) compared to normoxia (Fig. 4 & Table 1). Absence of any of the many MICOS complex components has been shown to reduce respiration rates in yeast (*Harner et al., 2011*), and in hypoxic human HEPG2 cells (*Plecitá-Hlavatá et al., 2016*). How MICOS protein levels are regulated in anoxia-tolerant species such as the crucian carp has not been investigated so far. We speculate that an opening of the cristae junction could lead to a decreased proton gradient, which then would stimulate substrate processing by the ETS complexes. Such a mechanism would also reduce the drive for ROS generation by RET but could also lead to reduced ATP synthesis by the ATP synthase. As mentioned, the energetic recovery from anoxia seems to be slower in the heart than in the brain and liver (*Dahl et al., 2021*), but the exact mechanism for this is still unknown. It could be that the decrease of MICOS proteins is part of a mechanism that suppresses oxidative phosphorylation within the first hours of reoxygenation. There are however indications that such a proton gradient does not form in the mitochondrial cristae (*Toth et al., 2020*), thus suggesting that this slow recovery of ATP levels in the heart could rather be due to reduced ETS enzyme levels.

Finally, FIS1 has recently been shown to have an essential role in the control of peripheral mitochondria fission, resulting in autophagy of small daughter mitochondria rather than apoptosis of the entire organelle (*Kleele et al., 2021*). Here, we observed an increase of hepatic FIS1 during anoxia-reoxygenation in liver (Fig. 5 & Table 1), but no differences in brain or heart. It is possible that FIS1-induced peripheral mitochondrial fission could be a mechanism used by crucian carp liver during anoxia-reoxygenation to minimize the amount of damaged cellular components.

Since accumulation of ROS is expected during reoxygenation and a major contributor to ischemia/reperfusion injury, we anticipated increased protein levels of antioxidant enzymes during anoxia and/or reoxygenation. However, proteins predicted to possess oxidoreductase activity were only over-represented among all significantly changed proteins in the liver (Fig. 3), where thioredoxin reductase 2 (TXNRD2) was among the proteins that exhibited the highest fold change increase during anoxia and reoxygenation compared to normoxia. It could be that anoxia-tolerant animals are constitutively equipped with high levels of ROS scavenging enzymes (*Storey, 1996*; *Hermes-Lima, Storey & Storey, 2001*), although this assumption has been questioned recently (*Bundgaard et al., 2019*; *Bundgaard et al., 2020*). The initial burst of mitochondrial superoxide has been demonstrated to be the main source of ROS in early phases of reperfusion in mammals, while activation of non-mitochondrial ROS production seems to occur at a later point during reperfusion (*Abramov, Scorziello & Duchen, 2007*; *Chouchani et al., 2016*). Thus, if mitochondrial ROS levels do not increase in the crucian carp at the onset of reoxygenation, it might suggest that ROS scavenging enzymes are present at sufficient levels and do not need to be upregulated. Absence of ROS production during reoxygenation in hypoxia-tolerant animals is also supported in naked mole-rats that do not experience increased ROS levels in the brain during reperfusion (*Eaton et al., 2022*).

## Glucose metabolism

Glycogen metabolism is obviously a key to anoxic survival in crucian carp, and most changes in protein levels were observed in the heart. Both glycogen debranching enzyme and glycogen synthesizing proteins (glycogen synthase and glycogenin) increased in anoxia and reoxygenation in the heart (Fig. 6 & Table 3). While increase of glycogen debranching enzyme during anoxia was expected, we did not expect this increase to continue into reoxygenation, nor did we expect glycogen-synthesizing proteins to increase in anoxia. However, it is possible that these changes function to prepare the tissue for the coming reoxygenation to allow a rapid build-up of depleted glycogen stores. Nevertheless, the increase of debranching enzyme (10-fold) was higher than the rise in glycogen synthesizing proteins (2–3-fold), consistent with the well-established role of glycogen catabolism during anoxia. This is in agreement with previous findings showing that cardiac glycogen is mobilized after one week of anoxia in crucian carp (*Vornanen & Haverinen, 2016*). Those authors observed the highest drop in glycogen content during anoxia in the liver, confirming the well-established role of hepatic glycogen catabolism for long-term anoxic survival in crucian carp (*Nilsson, 1990*). It was therefore surprising to see that the protein levels of neither glycogen phosphorylase, the key enzyme for glycogen breakdown, nor glycogen debranching enzymes were affected by anoxia in the liver (Fig. 6). Glycogen breakdown in the liver is seemingly not regulated by the amount of glycogen catabolic enzymes present. The only significantly changing glycogen-related protein in the liver was the regulatory subunit beta of phosphorylase b kinase (PHKB; Table 3) which increased in anoxia and reoxygenation. If the observed increase in PHKB protein levels also leads to increased phosphorylation of glycogen phosphorylase, our data suggest that the upregulation of PHKB may contribute to glycogen breakdown in the liver of crucian carp.

Surprisingly few glycolytic enzymes were changed during anoxia exposure, although we expect glycolytic rate to increase. The only tissue that showed increased levels of glycolysis-related proteins was the heart, where increased protein levels of the rate-limiting HK2 suggest elevated glucose flux into glycolysis (Fig. 6 & Table 3). This is in line with the observed upregulation of glycolytic genes in anoxic crucian carp heart (*Stensløkken et al., 2014*). Hypoxia-induced upregulation of HK2 has been shown to have an impact on various cell survival pathways during hypoxia and ischemia, acting on energy conservation, mitochondrial integrity protection and reduced ROS when oxygen returns (*Roberts & Miyamoto, 2015*), and it is possible that it also has a protective role in anoxia-tolerant vertebrates.

## Complement system

Activation of the complement system occurs in several mammalian tissues upon ischemia/reperfusion injury (*Arumugam et al., 2004*; *Eltzschig & Eckle, 2011*), but has so far not been investigated in anoxia-tolerant animals during oxygen depletion. In this current study, activation of the complement system was apparent in the heart tissue especially during anoxia (Fig. 3 & Table 4), but also to some extent during reoxygenation. Even though the liver is seen as the main producer of complement components (*Schartz & Tenner, 2020*), there is also evidence for local complement protein generation in other

tissues such as neuronal cells, T cells, cardiomyocytes and endothelial cells (*Yasojima et al., 1998*; *Emmens et al., 2016*; *Arbore, Kemper & Kolev, 2017*). We detected fewer complement components in the crucian carp liver, and those found were unaffected by oxygen levels, suggesting that the changes in amount of cardiac component proteins most likely have a particular role in the heart.

While ROS generation is a likely source for complement activation in reperfusion/reoxygenation (*Granger & Kvietys, 2015*), the anoxia-activated complement system in crucian carp heart cannot be explained by this mechanism. However, activation of the complement system has also been attained a role in removal of apoptotic cells (*Ricklin et al., 2010*), and in cardiac tissue repair in regenerative models (*Farache Trajano & Smart, 2021*). For example, complement signaling has been shown to be required for tissue regeneration and repair after myocardial injury in the zebrafish (*Natarajan et al., 2018*), including hypoxia/reoxygenation (*Parente et al., 2013*). We speculate that a similar mechanism can be present in the crucian carp, since goldfish heart also has been shown to regenerate after tissue injury (*Grivas et al., 2014*). Further studies are certainly required to fully understand the role of complement activation in the crucian carp.

## CONCLUSIONS

While most studies on crucian carp so far have focused on selected candidate proteins, we have applied a holistic, untargeted approach to reveal unknown adaptations to anoxia and reoxygenation in this study. We show that survival during anoxia and reoxygenation is controlled through regulation of protein levels of a set of key proteins and pathways rather than widespread changes of the entire global proteome. While keeping in mind that protein function often is controlled by several factors in addition to the regulation of protein content, our data point to tissue-specific adaptations to anoxia and reoxygenation in the crucian carp. Tissue-specific differences included mitochondrial proteins involved in succinate metabolism, electron transport and oxidative phosphorylation, many pointing towards adaptations to cope with oxygen depletion and ROS generation. Although we acknowledge the somewhat limited sample size used in this current study, the findings are statistically valid and suggest many exiting directions to follow up on in this remarkable model of anoxia tolerance.

## ACKNOWLEDGEMENTS

We thank Helge-André Dahl for practical assistance during tissue sampling, Margarita Strozynski for technical assistance in the lab and Gigi Lau for valuable input on the manuscript.

### Funding

This work was financially supported by the University of Oslo. Mass spectrometry-based proteomic analyses were performed by the Proteomics Core Facility, Department of

Biosciences, University of Oslo. This facility is a member of the National Network of Advanced Proteomics Infrastructure (NAPI), which is funded by the Research Council of Norway INFRASTRUKTUR-program (project number: 295910). The funders had no role in study design, data collection and analysis, decision to publish, or preparation of the manuscript.

### Grant Disclosures

The following grant information was disclosed by the authors:
University of Oslo.
Mass spectrometry-based proteomic analyses were performed by the Proteomics Core Facility, Department of Biosciences, University of Oslo.
National Network of Advanced Proteomics Infrastructure (NAPI), which is funded by the Research Council of Norway INFRASTRUKTUR-program: 295910.

### Competing Interests

The authors declare there are no competing interests.

### Author Contributions

- Anette Johansen conceived and designed the experiments, performed the experiments, analyzed the data, prepared figures and/or tables, authored or reviewed drafts of the article, and approved the final draft.
- Bernd Thiede conceived and designed the experiments, analyzed the data, authored or reviewed drafts of the article, and approved the final draft.
- Jan Haug Anonsen analyzed the data, authored or reviewed drafts of the article, and approved the final draft.
- Göran E. Nilsson conceived and designed the experiments, authored or reviewed drafts of the article, and approved the final draft.

### Animal Ethics

The following information was supplied relating to ethical approvals (*i.e.*, approving body and any reference numbers):

All experimental procedures were approved by the Norwegian Animal Health Authority (FOTS ID 16063), thereby following all relevant Norwegian and European Union guidelines and regulations.

### Data Availability

The mass spectrometry proteomics data are available at the ProteomeXchange Consortium *via* the PRIDE partner repository: http://dx.doi.org/10.6019/PXD032830.

The raw data from label-free quantitation, complete list of all regulated proteins, enrichment analysis and PCA loadings are available in the Supplementary Files.

### Supplemental Information

Supplemental information for this article can be found online at http://dx.doi.org/10.7717/peerj.14890#supplemental-information.

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
