# Peer review of "Surviving without oxygen involves major tissue specific changes in the proteome of crucian carp (Carassius carassius)"

_PeerJ, doi:10.7717/peerj.14890_

## Round 0.1 · original submission · Minor Revisions

Overall the reviewers found the study very interesting and commented on the thoroughness of this well-conducted study. They have highlighted some areas for improvement that I invite the authors to address. I would advise the authors pay particular attention to the reviewers' comments on data interpretation.

·

Basic reporting

See below

Experimental design

See below

Validity of the findings

See below

Additional comments

Overall comments: In this manuscript the authors explore the impact of prolonged anoxia and reoxygenation on the proteome of carp liver, heart, and brain. The study is well-conducted, and the manuscript is well-written. The authors have world-leading expertise in the study of this species and this study is an important addition to the volume of work they have contributed to the field of anoxia-tolerance over the past several decades. I particularly enjoyed the thorough presentation and discussion of the results. Whereas many studies of this type often gloss over the bulk of the changes by grouping them into pathways and key terms, I appreciated that the authors tried to describe the likely impact of their results at the level of the various tissues and in groupings of key metabolic pathways and how these impact the ecophysiology of the species. I have a few comments that I hope the authors will consider in revising their manuscript.


Major comments:

- I am not an expert in the analysis of proteomics data but I found the statement that results were “filtered to keep only proteins detected in minimum three replicates in at least one experimental group” to be somewhat concerning. Does this imply that for some experimental groups only 1 or 2 samples contained the protein of interest? If so, applying an average area threshold to fill in the missing values may significantly bias the outcome. Can the authors indicate which of their proteins of interest did not have hits in at least 3 samples in each treatment group for reasonable comparison? For example, the authors state that TOM40 was only found in normoxia. Does this imply that it is entirely absent in anoxia and during reoxygenation? Or that you were unable to detect it for some technical reason? This is an important distinction.


Minor comments:

- The title is a bit sweeping as it implies that all anoxia tolerant species have major tissue-specific changes in their proteome. Please make the title specific to the study species.

- Abstract: the carp is not the only species that can survive months of anoxia at low temperatures and thus it is not a “unique” model for studying molecular adaptations to hypoxia. Please change to “excellent” or another adjective that is more appropriate.

- Language choice: The authors often state that proteins were “not regulated” or “regulated” in various tissues but I think what they mean to say is that the expression of these proteins did not change or was not impacted, etc. Certainly, the expression of proteins is always regulated, even if they do not change. Indeed, if they were not regulated, then their expression would be chaotic, and we might expect large changes!

- The manuscript is generally well-written but there are numerous grammatical and syntax errors and I encourage the authors to proof-read their submission more carefully. Some examples (there are several others):

o Abstract: change to “how its global preteome responds to…”
o Abstract: The results emphasizes (plural agreement issue)
o Results: To investigate how the proteome of carp changes with oxygen…
o Results: line 203, start line with “as”
o Line 204: “according to …”
o Line 245 “depite of the”…delete of
o Line 380 “we like to propose” change to “we propose”
o Line 343 “a lot of attention”
o Line 410 and line 411: ETS and ETC are used…please be consistent. Also line 464 and line 465.
o Line 476: by the ATP synthase
o Line 578: somewhat instead of somehow…the how is your choice of sample size!

- Introduction first sentence: I am not aware of studies showing that goldfish can survive anoxia for months. Certainly a few days but not months. This is a statement that is often made about goldfish by the authors but then citations are provided to work in carp. Please provide citations of studies in which goldfish have been shown capable of surviving months of anoxia.

- Please provide an animal ethics statement and standard details about the ethics body that approved this study.

- Line 341 – citation is in a different format here.

- Lines 507-509: We have also seen high levels of endogenous glutathione-related scavenging capacity in naked mole-rat heart (See Munro et al Aging Cell 2019), thus the distinction may be between hypoxia-tolerant and intolerant species as opposed to reptiles/amphibians vs. mammals. Similarly, we do not see an increase in ROS with reperfusion in naked mole-rats (Eaton et al, Current Neuropharm 2022)…so again, the distinction vs. mammals is too broad.


Matthew Pamenter

Reviewer 2 ·

Basic reporting

This paper presents a throughout and careful analysis of the proteomic changes following anoxia and reoxygenation in crucian carp heart, liver and brain, showing remarkable and unexpected tissue-specific differences. The authors are experts in anoxia tolerance molecular and biochemical mechanisms in crucian carp and provide in this paper important insights into such mechanisms in this iconic species. This work is also a valuable follow-up of a recent metabolomic study on crucian carp published by the authors and made under identical conditions as the present study, allowing interpretation of results into a particularly solid framework compared to other standalone proteomic studies.
The paper is well-written and the experimental work is solid and well presented. To further improve clarity and impact, the authors may consider expanding the introduction (which is rather short) and reducing the discussion (which is rather long) by moving some background information from the discussion section into the introduction and by sparing some speculative parts (those on antioxidants and on cristae formation, for example). Also, formulating hypotheses based on previous studies – including the authors’ own metabolomic work - in the introduction will improve readability along the paper, in my opinion, which is rather dense in terms of results and their detailed interpretation. This would give more space to discuss the main findings.

Experimental design

no comment.

Validity of the findings

conclusion could be expanded to include what are the main regulations in the three organs as a sort of take-home message to wrap up the paper. I would like to stress that these comments are meant to be suggestions to improve the impact of such interesting results. I have no concern on the quality of the data.

Additional comments

Specific points:
Line 96: replace mixed with both
Line 109: please clarify if tissues were rinsed to remove blood
Figure 2, line 211: please state in figure legend what determines the size of the circles and insert number to improve clarity. Numbers in the main text cannot be found in the figure
Line 227: although it refers to the method used, the term ‘enrichment’ in this context could be misleading, especially because a ‘mitochondrial enriched fraction’ is a semi-purification of mitochondria. I wonder whether this part could be reformulated to clarify that these are the pathways identified by the method
Line 248-249: In the main text, it would be better to use the same category notation used in figure 3, to improve clarity
Line 277: please insert reference to Table 2 after parentheses. Also, I would suggest using the same number of decimals in the main text as in the Table, to ease cross reference. In Table 2, what exactly is the p value in column? Does it refer to any of the two ratios?
Line 292: The issue of ATP synthase inhibition in the heart is interesting, perhaps this finding would deserve some more space in the discussion

Reviewer 3 ·

Basic reporting

I have a few minor comments for the authors to consider as they finalize their manuscript.

There can be a tendency to over interpret or over extrapolate the functional consequences of omics-generated data. I generally applaud the authors for appropriately discussing their data in this context, which of course is made easier by looking at changes in protein content. The authors do, however, periodically over interpret the significance of their results. One example is related to the discussion glucose metabolism (around line 524) where differences in the amount that glycogen debranching enzyme increased compared with enzymes associated with glycogen synthase, which was used to make arguments about the relative importance of pathway (“indicating that glycogen catabolism is the predominant pathway during anoxia in the heart“). Changes in protein amount do not necessary result in changes in metabolic flux. In this particular case, I don’t necessarily disagree with the conclusion that glycogen catabolism dominates during anoxia, but my agreement is based upon knowledge of the study system, rather than the presented protein amounts. It might be better to use language like… the large increase in debranching enzyme is consistent with the well-established role of glycogen during anoxia….” Or something similar. This issue applies in this example and elsewhere, so I suggest the authors review their manuscript with this in mind.

In the abstract, and throughout the MS, the terms downregulated and upregulated are used to describe decreases and increases in the amount of specific proteins, which is common; however, in the abstract the following statement is made “but not regulated in the brain”. A lack of change in protein content does not mean the amount of protein expressed is not under tight regulation, which draws into questions the use of “regulation” when describing increases and decreases. Consider changing to decrease, increase and no change in protein amount. The text on line 210-211 also illustrates this issue.

Line 66. “were” instead of “was”

Figure 2. The venn diagram shown in Figure 2 could include more information, including the number of proteins that differentially expressed only in one tissue and possibly a text breakdown of those showing increases versus decreases. Almost all the data described in the text from lines 210 to 221 could be nicely encapsulated in Figure 2.

Line 338 consider replacing overlapping with overlap
Line 380 consider “We would like…”
Line 403 consider “decrease(s)”

Experimental design

No concerns. Proteomics study in multiple tissues from the Crucian carp in normoxia, after anoxia exposure, and after anoxia and recovery in normoxia. Methodology is sound and supplemental material appears to contain all the relevant data to support the results and conclusions.

Validity of the findings

No comment

---

## Round 0.2 · accepted · Accept

The authors were able to address the reviewers' comments in their revised manuscript. I have reviewed these changes and feel the manuscript is now ready for publication.